# Trends of *Mycobacterium tuberculosis* and rifampicin resistance at the Ho Teaching Hospital in Ghana

**John Gameli Deku**[1]*, **Enoch Aninagyei**[2], **Israel Bedzina**[3], **Gameli Nudo**[4], **Emmanuel Ativi**[4], **Prosper Mensah**[4], **Solomon Wireko**[5], **Aaron Osei-Tutu**[1], **Emmanuel Duker**[1], **Innocent Afeke**[1]

1 Department of Medical Laboratory Sciences, School of Allied Health Sciences, University of Health and Allied Sciences, Ho, Ghana, 2 Department of Biomedical Sciences, School of Basic and Biomedical Sciences, University of Health and Allied Sciences, Ho, Ghana, 3 Reinbee Medical Laboratory and Wellness Center, Ho, Ghana, 4 Laboratory Department, Ho Teaching Hospital, Ho, Ghana, 5 Department of Laboratory Technology, Kumasi Technical University, Kumasi, Ghana

* sssdeku@gmail.com

**Data Availability Statement:** All relevant data have been uploaded as a supporting file.

**Funding:** The author(s) received no specific funding for this work.

## Abstract

### Background

Tuberculosis remains a major public health threat worldwide, causing significant morbidity and mortality, particularly in low- and middle-income countries. In recent years, efforts to combat tuberculosis have focused on strengthening healthcare systems and increasing access to diagnostics and treatment services. There is scarcity of data on the prevalence of *Mycobacterium tuberculosis* and rifampicin-resistant tuberculosis in the Volta region of Ghana. Therefore, the aim of this study was to determine the trends of *Mycobacterium tuberculosis* and rifampicin resistance in a major teaching hospital in Ghana spanning a six-year period.

### Methodology

A retrospective cross-sectional hospital study was conducted at Ho Teaching Hospital, Ho, Ghana. Study data included archived results on tuberculosis testing using GeneXpert from 2016–2021. Archived data on tuberculosis testing were collected and entered using Microsoft Excel 2019. IBM SPSS (v26) was used for a statistical analysis of the prevalence of tuberculosis. P-value <0.05 was considered statistically significant.

### Results

The study included 5128 presumptive tuberculosis cases from 2016 to 2021, of which 552 were positive, revealing an overall prevalence of 10.76%. Males exhibited a significantly higher prevalence of tuberculosis (14.20%) compared to females (7.48%), with a male-to-female ratio of 2:1. The burden of tuberculosis varied significantly between age groups, with those aged 30–45 years and 46–60 years facing twice the risk compared to those under 15 years (p<0.001). Rainy seasons correlated with heightened tuberculosis occurrences

**Competing interests:** The authors declared no conflict of interest.

(12.12%) compared to dry seasons (8.84%) (p = 0.008). Rifampicin-resistant tuberculosis was prevalent at 3.45%, slightly higher in women, particularly in the 45–59 age group (5.97%). In particular, tuberculosis prevalence exhibited fluctuations, peaking in 2016 (17.1%) and 2020 (11.5%), with a trough in 2019 (4.6%).

## Conclusion

The overall prevalence of laboratory confirmed tuberculosis was 10.76%, and resistance to rifampicin, 3.45%, indicating high infection and possible treatment failure. Considering its infectious nature, this calls for concerted efforts to curb the spread of the infection.

## Introduction

Tuberculosis, which is caused by bacteria of the *Mycobacterium tuberculosis (M. tuberculosis* complex), is one of the oldest diseases known to affect humans and a major cause of death worldwide [1]. *Mycobacterium africanum (M. africanum), Mycobacterium canetti (M. canetti), Mycobacterium caprae (M. caprae), Mycobacterium microti (M. microti),* and *Mycobacterium pinnipedii (M. pinnipedii)* are responsible for a negligible portion of human cases. People with pulmonary tuberculosis emit aerosolized germs, especially when coughing, causing the disease to spread via the air [2].

A third of the globe is believed to be infected with *M. tuberculosis* but it is asymptomatic, which is described as having latent tuberculosis, despite the fact that 9 million new cases of active tuberculosis are recorded each year [3]. The World Health Organization estimates that of the more than 9 million people who contract tuberculosis each year, more than 3 million do not receive a standard diagnosis, treatment course, or registration by national tuberculosis programmes [4].

Tuberculosis remains a major public health problem in Ghana. The two mainstay medications of the directly observed therapy short-course (DOTS) regimen, isoniazid (INH) and rifampicin (RIF), can cause drug resistance, which can have a detrimental effect on the effectiveness of tuberculosis treatment [5]. A study by Owusu-Dabo, Adjei [6], reported an overall primary drug resistant rate of 23.5% in Ghanaian patients with tuberculosis. In a national drug resistance survey among tuberculosis patients in Ghana by Sylverken, Kwarteng [7], drug resistance was reported to be 25.2%, with 2.4% resistance to RIF. Other studies done in eight out of the sixteen regions of Ghana, and in Accra reported the prevalence of RIF resistance to be 16.8% [8] and 30.4% [9] respectively.

Policy makers can use the data from tuberculosis prevalence surveys to enhance reporting systems, as well as tuberculosis diagnosis and treatment programmes. To curb the spread of tuberculosis and reduce its impact, early diagnosis and treatment are essential. However, conducting routine tuberculosis prevalence surveys is expensive and logistically challenging [10]. This notwithstanding, countries such as Lesotho [11] and Myanmar [12] have been successful.

The resurgence of tuberculosis as a global concern was caused by the lack of an effective vaccine, the long and expensive drug regimens, the limited diagnostic tools available in endemic countries, and the dismantling in several nations of the health systems and control measures that successfully contributed to control of tuberculosis throughout most of the 20th century [13].

Although there has been some progress with the treatment and elimination of tuberculosis in Ghana, there is a scarcity of data on the prevalence and rifampicin-resistant tuberculosis in

the Volta Region of Ghana. This study therefore determined the prevalence of tuberculosis and its resistance to rifampicin over a six-year period (2016–2021) at the Ho Teaching Hospital, the major referral health centre in the Volta region.

## Materials and methods

### Study design

The study was a hospital-based retrospective study that analysed secondary data collected on tuberculosis from the Microbiology Laboratory Unit of the Ho Teaching Hospital (HTH). Data analysed in this publication were archived from 2016–2021.

### Study site

The study was carried out at the HTH microbiology laboratory unit in the Ho Municipality, Volta Region, Ghana. This laboratory unit is responsible for the processing of urine, stool, sputum, and other bodily fluids for infectious pathogens. Ho, the capital city of the Volta region, is located between Mount Adaklu and Mount Galenukui. According to the 2010 Population and Housing Census, the population of Ho municipality is approximately 177,281, constituting 8.4% of the region's total population. Females make up 52.7% of the population. About 62% of the population resides in urban localities [14]. The municipality shares boundaries with the Adaklu and Agotime-Ziope districts to the south, the Ho West district to the north and west, and the Republic of Togo to the east. Its total land area is 2361 square kilometers (912 sq. km), representing 1.5% of the region's total land area. The coordinates of HTH are 6.60126˚N 0.48404˚E. HTH serves as the main referral facility in the Volta region, serving over 100,000 patients and boasting a bed capacity of 340.

### Study participants

The study participants included records of tuberculosis test performed and archived between 2016 and 2021 at the Ho Teaching Hospital.

### Inclusion and exclusion criteria

All tuberculosis test records performed and archived between 2016 and 2021 were included in the study. However, tuberculosis test records with incomplete data (missing age, gender, and test results) were excluded from the study.

### Laboratory testing (GeneXpert assay)

Sputum samples with sufficient volume were tested with the Xpert *Mycobacterium tuberculosis*/Rifampicin (MTB/RIF) version 5.0 assay on the GeneXpert platform (Cepheid, USA) to detect the presence of *Mycobacterium tuberculosis* and rifampicin resistance, according to the manufacturer's instructions.

### Data collection

Archived hospital data were collected from the hospital microbiology unit, sorted and entered into Microsoft Excel 2019 file and prepared for analysis. The data recovered were age, gender, year of diagnosis, and result of the tuberculosis test. The archived records were accessed between October 3, 2022, and October 22, 2022.

## Outcome variables

The outcome variable for this study was the prevalent rate of tuberculosis cases, the prevalence of rifampicin resistance among tuberculosis cases, and the overall trends of rifampicin resistance during the study period.

## Independent variables

The years used in the study, age and gender of the patients were independent variables in this study.

## Data analysis

Data collected with Microsoft Excel 2019 were exported into IBM Statistical Package for the Social Sciences version 26.0 (SPSS v.26) later for statistical analysis. Descriptive statistics were conducted to analyze the demographics of the participants and the trends in the prevalence of tuberculosis during the study period. A logistic regression analysis was used to determine predictors of tuberculosis among presumptive cases. A *P value* <0.05 was considered to indicate statistical significance.

## Ethical consideration

Ethical approval with reference number UHAS-REC A.1 [139] 22–23 was obtained from the Research Ethics Committee of the University of Health and Allied Sciences. Additionally, written permission was sought from the management of the Ho Teaching Hospital for the use of data generated in the microbiology unit at the facility for the study. Informed consent was not obtained from participants since the study was retrospective. No information that could be linked to the participants was retrieved. All archived data for the study was kept undisclosed and used for the study only by keeping them on a password-protected computer accessible to only the principal investigator.

## Results

A total of 5128 presumptive tuberculosis cases were received for laboratory confirmation within the period between 2016 and 2021. Of this, 2507 (48.89%) were males and the remaining 2621 (51.11%) were females. The overall prevalence of laboratory confirmed tuberculosis was 10.76% among presumptive cases. The prevalence of laboratory confirmed tuberculosis was significantly higher among males (14.20%) than their female counterpart (7.48%) with male-to-female ratio of 2:1 (p<0.001). After adjusting for confounding variables, males were twice more likely to be burdened with tuberculosis than their female counterparts (aOR2.07 (1.72–2.49). The burden of laboratory confirmed tuberculosis varied significantly between various age classes (p < 0.001) with persons under 15 years of age being the least burdened. The highest risk of tuberculosis was among the middle aged [30–45 years (13.35%)]. The age groups 30–45 years and 46–60 years were twice as likely to have tuberculosis compared to those under 15 years [(aOR = 2.74, 95% CI: 1.74–4.31, p<0.01) and (aOR = 2.58, 95% CI: 1.64–34.07, p<0.01) respectively]. Also, climatic seasons were significantly associated with the appearance of tuberculosis infection (p<0.01). Tuberculosis cases were identified more frequently in the rainy season [365(12.12%), aOR = 1.30, p<0.01] than in the dry season [187 (8.84%)]. Table 1.

The overall prevalence of newly diagnosed rifampicin-resistant tuberculosis (RR-TB) was 3.45%. The prevalence of RR-TB was comparably higher among females than males (4.00% vs. 3.16%, respectively, p = 0.395). Similarly, a comparable prevalence of RR-TB was observed

**Table 1. Prevalence and factors of tuberculosis among suspected cases at the Ho Teaching Hospital, 2016–2021.**

| Characteristics | Total | MTB Detected | P-value | cOR (95% CI) | P-value | aOR (95% CI) | P-value |
|---|---|---|---|---|---|---|---|
| Overall | 5128 | 552(10.76) | | | | | |
| **Age Group** | | | | | | | |
| <15 years | 385 | 23(5.97) | <0.001 | 1 | | 1 | |
| 15–29 years | 907 | 86(9.48) | | 1.65(1.02–2.65) | 0.040 | 1.92 (1.19–3.11) | **0.008** |
| 30–44 years | 1326 | 177(13.35) | | 2.42(1.55–3.80) | <0.001 | 2.74 (1.74–4.31) | <**0.001** |
| 45–59 years | 1292 | 167(12.93) | | 2.34(1.49–3.67) | <0.001 | 2.58 (1.64–4.07) | <**0.001** |
| >/ = 60 years | 1218 | 99(8.13) | | 1.39(0.87–2.23) | 0.166 | 1.53 (0.95–2.45) | 0.079 |
| **Gender** | | | | | | | |
| Female | 2621 | 196(7.48) | <0.001 | 1 | | 1 | |
| Male | 2507 | 356(14.20) | | 2.05(1.70–2.46) | <0.001 | 2.07 (1.72–2.49) | <**0.001** |
| **YEAR for TB infection** | | | | | | | |
| 2016 | 252 | 43(17.06) | <0.001 | 1 | | 1 | |
| 2017 | 1158 | 144(12.44) | | 0.69(0.48–1.00) | 0.051 | 0.71 (0.49–1.04) | 0.080 |
| 2018 | 1444 | 139(9.63) | | 0.52(0.36–0.75) | 0.001 | 0.55 (0.38–0.80) | **0.002** |
| 2019 | 241 | 11(4.56) | | 0.23(0.12–0.46) | <0.001 | 0.27 (0.13–0.55) | <**0.001** |
| 2020 | 1019 | 117(11.48) | | 0.63(0.43–0.92) | 0.018 | 0.66 (0.45–0.97) | **0.036** |
| 2021 | 1014 | 98(9.66) | | 0.52(0.35–0.77) | 0.001 | 0.56 (0.37–0.83) | **0.004** |
| **Season** | | | | | | | |
| Dry Season | 2116 | 187(8.84) | <0.001 | 1 | | 1 | |
| Rainy Season | 3012 | 365(12.12) | | 1.42(1.18–1.71) | <0.001 | 1.30 (1.07–1.58) | **0.008** |
| **Quarter of the Year** | | | | | | | |
| First Quarter | 1244 | 132(10.61) | <0.001 | 1 | | | |
| Second Quarter | 1002 | 122(12.18) | | 1.17(0.90–1.52) | 0.245 | | |
| Third Quarter | 1193 | 156(13.08) | | 1.27(0.99–1.62) | 0.060 | | |
| Fourth Quarter | 1689 | 142(8.41) | | 0.77(0.60–0.99) | 0.043 | | |

cOR- crude odds ratio, aOR- adjusted odds ratio, CI-confidence interval, MTB-*Mycobacterium tuberculosis*

between the various age classes, with the patient 45–59 years having the highest resistance burden with a prevalence of 5.97% (p = 0.627). None of the patients under 15 years of age was diagnosed with RR-TB. Furthermore, the burden of newly diagnosed RR-TB varied significantly during the review period (p = 0.038). There was an oscillating trend in the prevalence of RR-TB throughout the period from 2016 (4.65%) to 2017 (2.80%), 2018 (4.32%), 2019 (0.00%) and 2020 (5.88%) to 2021 (1.43%). However, seasonal variations were not associated with the appearance of resistance to rifampicin among newly diagnosed tuberculosis cases (p = 0.738). Table 2

In each age group, there were more males than females that were burdened with tuberculosis. However, while the male and female burden of tuberculosis was comparable between the age group of less than 15 years and those between 15 and 29 years, a significantly higher infection burden was observed between the males than females for the age groups 30–44 years, 45 to 59 years, and ≥ 60 years, as shown in Fig 1.

The cumulative burden of pulmonary tuberculosis within the review period was observed in the month of September (15.7%). However, among males, cumulative tuberculosis cases were detected most frequently among presumptive cases in May, and, cumulatively, positivity to tuberculosis was least observed in October and November. While among males, tuberculosis positivity among presumptive cases was least reported in November, October was the month

**Table 2. Prevalence and risk factors of newly diagnosed rifampicin resistant tuberculosis among suspected cases at the Ho Teaching Hospital between 2016 and 2021.**

| Risk factor | Total | Susceptible | Indeterminate | Resistant | P-value |
|---|---|---|---|---|---|
| Overall | 435 | 413(94.94) | 7(1.61) | 15(3.45) | |
| **Age Group** | | | | | |
| <15 years | 19 | 19(100.00) | 0(0.00) | 0(0.00) | 0.627 |
| 15–29 years | 73 | 71(97.26) | 1(1.37) | 1(1.37) | |
| 30–44 years | 128 | 121(94.53) | 2(1.56) | 5(3.91) | |
| 45–59 years | 134 | 124(92.54) | 2(1.49) | 8(5.97) | |
| >/ = 60 years | 81 | 78(96.30) | 2(2.47) | 1(1.23) | |
| **Gender** | | | | | |
| Female | 150 | 140(93.33) | 4(2.67) | 6(4.00) | 0.395 |
| Male | 285 | 273(95.79) | 3(1.05) | 9(3.16) | |
| **YEAR** | | | | | |
| 2016 | 43 | 41(95.35) | 0(0.00) | 2(4.65) | **0.038** |
| 2017 | 143 | 135(94.41) | 4(2.80) | 4(2.80) | |
| 2018 | 139 | 133(95.68) | 0(0.00) | 6(4.32) | |
| 2019 | 6 | 6(100.00) | 0(0.00) | 0(0.00) | |
| 2020 | 34 | 29(85.29) | 3(8.82) | 2(5.88) | |
| 2021 | 70 | 69(98.57) | 0(0.00) | 1(1.43) | |
| **Season** | | | | | |
| Rainy season | 302 | 287(95.03) | 4(1.32) | 11(3.64) | 0.738 |
| Dry season | 133 | 126(94.74) | 3(2.26) | 4(3.01) | |

with the least burden of tuberculosis positivity among females. The monthly observed variations in tuberculosis detection were statistically significant.

A comparable burden of tuberculosis was observed between presumptive cases of male and female in the months between June and September (p>0.05). In contrast, from October and March, a significantly higher burden of *Mycobacterium tuberculosis* was detected among the male presumptive compared to females (p<0.05). Fig 2

There was a significant trend in tuberculosis prevalence over the period, with a decrease in the prevalence from a peak in 2016 (17.1%) to 2019 (4.6%) where there was a trough in the prevalence. However, there was a sharp increase in the tuberculosis rate from 4.6% in 2019 to 11.5% in 2020 with a slight decline in 2021 (9.7%) (p-value$_{(trend)}$ = 0.016). Although not significant, a similar trend of tuberculosis infection was also observed among males and female when tuberculosis was the most prevalent in 2016 and the least prevalent in 2019, as depicted in Fig 3.

## Discussion

The retrospective study conducted over six years at the Ho Teaching Hospital sheds light on the trends of *Mycobacterium tuberculosis* and resistance to rifampicin among presumptive cases. A total of 5128 presumptive cases of tuberculosis were examined, 48.89% being males and 51.11% being females. The overall prevalence of laboratory confirmed tuberculosis was 10.76%, indicating a significant burden within the hospital's catchment area. Similar study by Arega, Menbere [15] in Addis Ababa and Boakye-Appiah, Steinmetz [16] in Komfo Anokye Teaching Hospital, Ghana reported a prevalence of 15.11% and 31.4%, respectively. The high prevalence reported by Boakye-Appiah, Steinmetz [16] for tuberculosis could be due to the small number of study population (376).

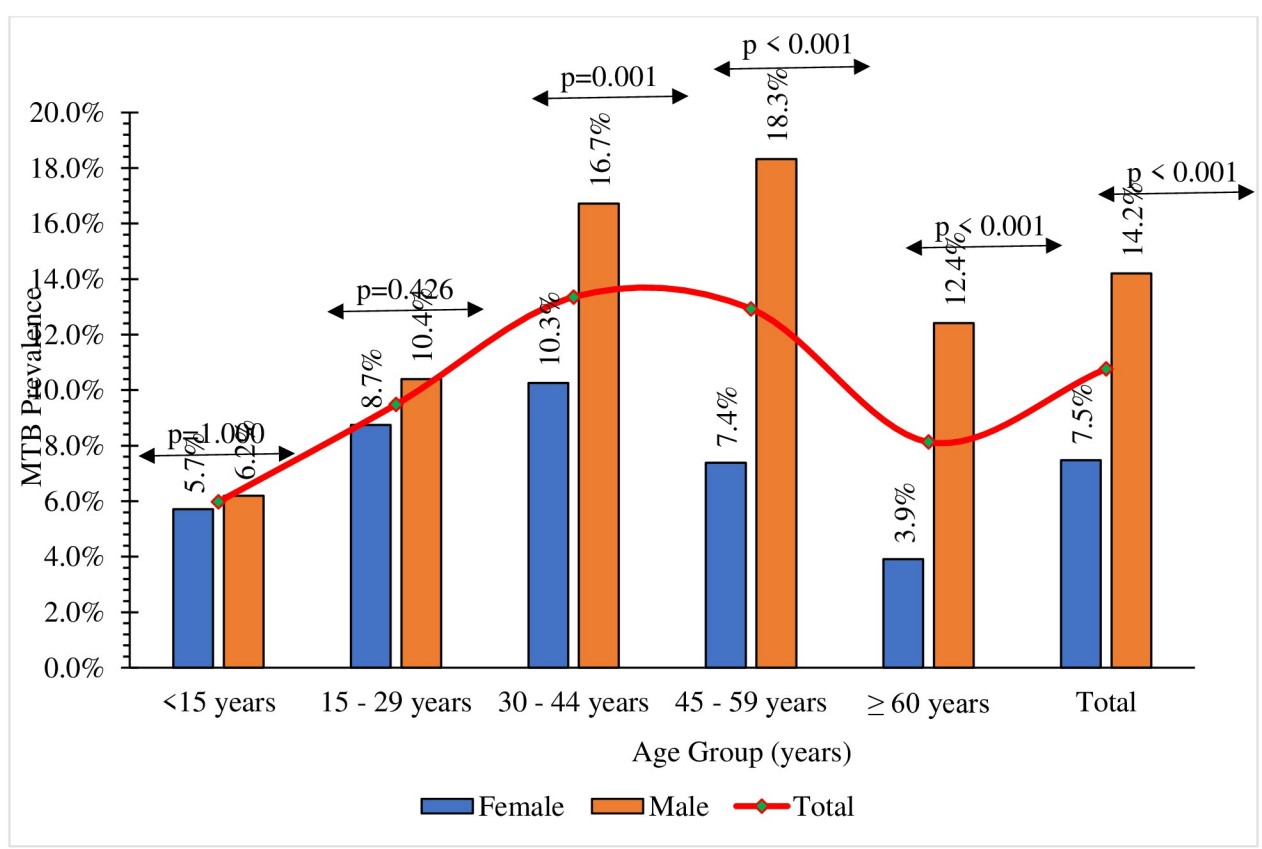

**Fig 1. Gender and variations in the prevalence of tuberculosis between age groups.**

The prevalence was significantly higher among males at 14.20% compared to females at 7.48%, revealing a distinct gender disparity in tuberculosis rates. This finding is supported by the 2:1 male to female ratio, which means that males were at a significantly higher risk of being diagnosed with tuberculosis compared to their female counterparts. Our study is consistent with studies by Gallant, Duvvuri [17], Lisdawati, Puspandari [18], Moyo, Ismail [19] and Sinshaw, Kebede [4].

Furthermore, the study's adjusted odds ratio of 2.07 (95% CI: 1.72–2.49) further emphasizes the increased probability of males being burdened with tuberculosis compared to females, even after accounting for possible confounding variables. This gender-based disparity in tuberculosis prevalence suggests a possible underlying biological or behavioral vulnerability among males, underscoring the need for targeted interventions and awareness campaigns tailored to the specific challenges faced by this demographic.

In addition to gender-based differences, the burden of laboratory-confirmed tuberculosis exhibited substantial variations between different age groups. In particular, individuals under 15 years of age demonstrated the lowest prevalence, indicating a relatively lower susceptibility to tuberculosis within this cohort. However, the middle-aged group (30–45 years) exhibited the highest risk, with a prevalence of 13.35%. The adjusted odds ratios for the age groups 30–45 years and 46–60 years stood at 2.74 (95% CI 1.74–4.31) and 2.58 (95% CI 1.64–34.07) respectively, indicating a significantly elevated probability of tuberculosis occurrence compared to those under 15 years. This is consistent with a study by Kapata, Chanda-Kapata [20].

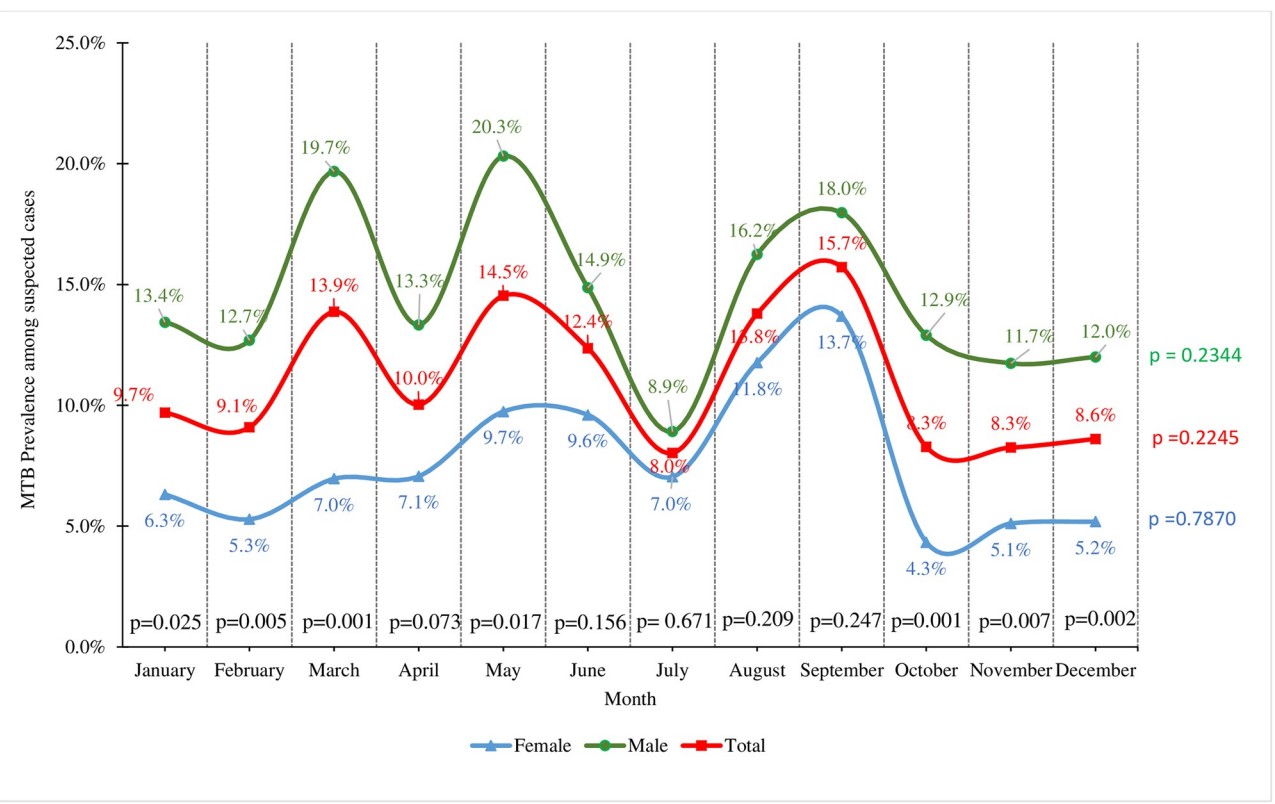

**Fig 2. Monthly variation in tuberculosis detection at the Ho Teaching Hospital, 2016–2021.**

Furthermore, the study highlighted the association between climatic seasons and tuberculosis occurrence, indicating a significant link between the prevalence of tuberculosis and the climatic conditions in the area. The identification of tuberculosis cases was notably more frequent during the rainy season, accounting for 12.12% of cases, compared to 8.84% during the dry season. This observation underscores the potential impact of environmental factors on the transmission dynamics of tuberculosis and emphasizes the need for targeted surveillance and preventive measures during specific climatic periods, especially during times of increased rainfall. These findings underscore the complex interplay of biological, demographic and environmental factors that influence the prevalence of tuberculosis. They emphasize the need for customized public health interventions and comprehensive strategies to address specific vulnerabilities identified among different demographic groups and during specific climatic seasons. This comprehensive approach is crucial to mitigating the burden of tuberculosis and improving overall health outcomes within the hospital's area of responsibility.

The findings of the retrospective study conducted at the Ho Teaching Hospital revealed several important patterns regarding the prevalence of newly diagnosed RR-TB. The study found an overall prevalence rate of 3.45% for RR-TB, indicating that a significant proportion of tuberculosis cases in the region are resistant to this crucial antibiotic. Other studies in Ghana by Boakye-Appiah, Steinmetz [16] and Sylverken, Kwarteng [7] in a national survey and Asante-Poku, Otchere [21] reported a RIF resistance of 14.4% and 2.4% and 1.5% respectively. Arega, Menbere [15] reported prevalence of RR-MTB in a study among presumptive tuberculosis patients in selected government hospital in Addis Ababa, Ethiopia to be 9.9%.

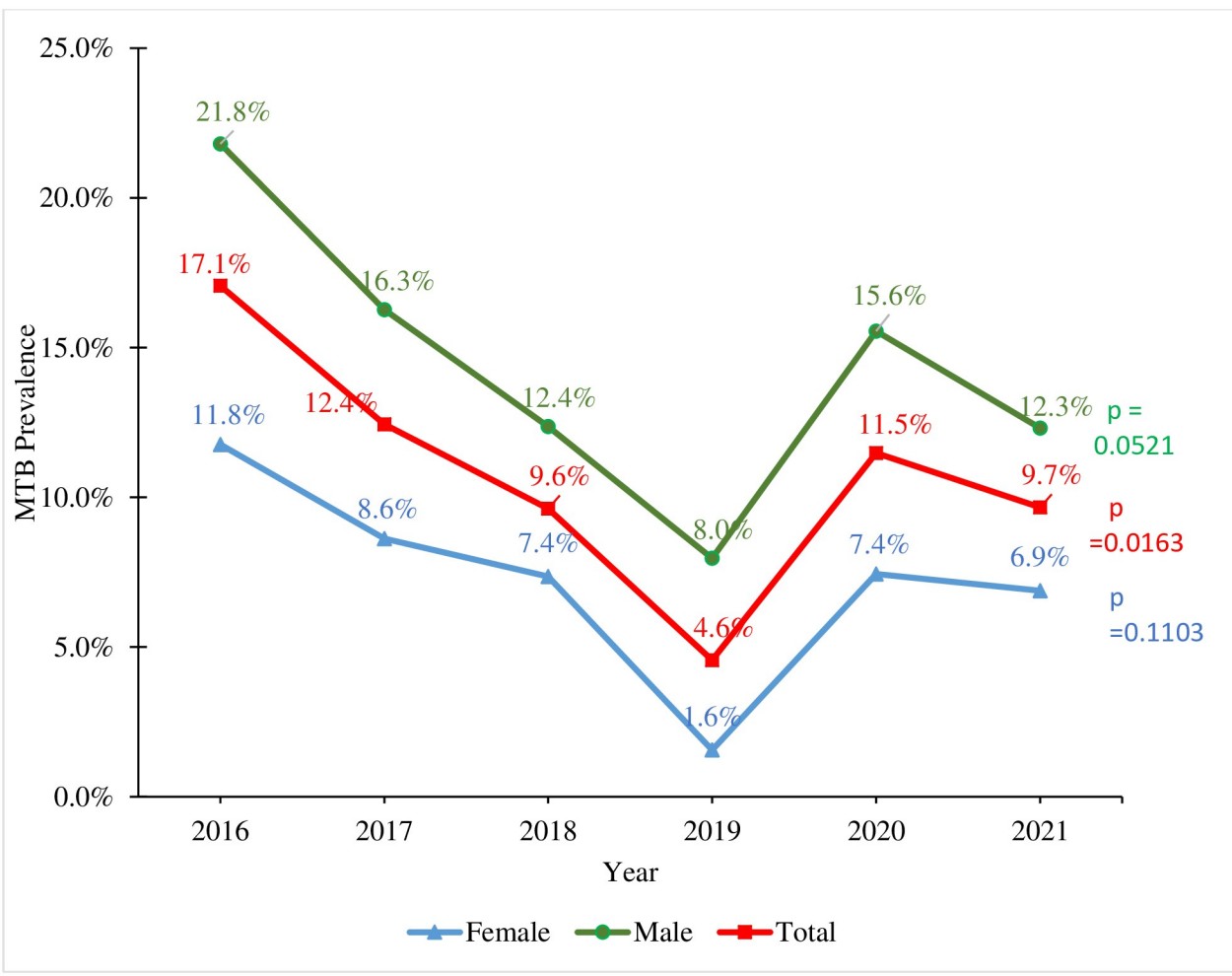

**Fig 3. Year-on-year trend of TB prevalence stratified by gender at the Ho Teaching Hospital.**

Surprisingly, the prevalence of RR-TB showed a slight predominance among females compared to males, with rates of 4.00% and 3.16%, respectively, although the observed difference was not statistically significant (p>0.05). This suggests that gender might not be a significant determinant of the appearance of RR-TB in this population. Although our study did not find statistical significance between male and females, being female was identified as an independent risk factor for RR-TB in a study by Arega, Menbere [15] and this may be related to socio-economic factors, probably due to lack of control of financial resources at household levels and poor knowledge about tuberculosis. In contrast, Ulasi, Nwachukwu [22] reported a higher prevalence among males in Enugu, Nigeria.

Furthermore, the study indicated that the age distribution of RR-TB cases did not show a stark variation, with comparable prevalence rates between different age groups. However, a notable observation was the relatively higher resistance burden among the 45–59 age group, with a prevalence of 5.97%. In particular, no cases of RR-TB were identified among patients under 15 years of age, implying a potential age-related factor in the resistance pattern. This trend could suggest a certain level of vulnerability among middle-aged individuals, indicating

the need for targeted interventions in this specific demographic group. In contrast to our finding, studies conducted in other settings reported higher RR-TB in ages below 45 years [22–24].

Interestingly, the study highlighted a significant fluctuation in the burden of newly diagnosed RR-TB over the six-year period under review (p<0.05). The prevalence of RR-TB exhibited an oscillating trend over the years, showing a pattern of inconsistency rather than a steady increase or decrease. This fluctuation was evident from the observed rates over the years, starting at 4.65% in 2016, followed by a decrease in 2017 to 2.80%, an increase in 2018 to 4.32%, a complete absence in 2019, a substantial increase in 2020 to 5.88%, and a subsequent decrease to 1.43% in 2021. These variations suggest the presence of potential contributing factors that could have influenced the dynamics of RR-TB prevalence during the study period.

On the contrary, the study results of the study indicated no significant association between climatic seasons and the occurrence of rifampicin resistance among the newly diagnosed tuberculosis cases (p>0.05). This finding suggests that the prevalence of RR-TB remained consistent throughout different seasons, implying that environmental factors, specifically those related to climate variations, may not be significant drivers of the prevalence of RR-TB within the study area.

The study provides critical information on into the prevalence of rifampicin-resistant tuberculosis within the Ho Teaching Hospital, highlighting the need for more research to elucidate the underlying factors driving the observed trends. Furthermore, the study underscores the importance of continuous monitoring and surveillance of RR-TB to implement targeted interventions and policies that can effectively combat the challenges posed by antibiotic resistance in tuberculosis management.

The results of this study indicate a notable prevalence of tuberculosis cases among males across all age groups, suggesting a potential gender-based disparity in the burden of the disease. Surprisingly, the study reveals a relatively balanced distribution of tuberculosis between males and females in age groups under 15 years and between 15 and 29 years. This balanced representation in the younger age cohorts may indicate a comparable vulnerability to tuberculosis infection in both genders, possibly due to similar exposure patterns and physiological factors influencing susceptibility during these stages of life.

However, a significant deviation from this trend appears in the subsequent age categories, 30–44 years, 45–59 years and 60 years and above. In these age groups, the burden of tuberculosis demonstrates a strikingly higher incidence among males compared to females. This disparity suggests a possible interplay of multiple factors, including disparities in access to healthcare, occupational hazards, behavioral patterns, and hormonal influences, which could contribute to the increased vulnerability of males to tuberculosis within these older age groups.

The observed gender-based differences in tuberculosis prevalence, particularly in the older age groups, point to the need for a more nuanced understanding of the social, biological and environmental determinants that contribute to the unequal distribution of tuberculosis burden. Furthermore, it emphasizes the importance of implementing targeted interventions tailored to address the specific risk factors and challenges faced by males in the older age cohorts, such as improved access to healthcare services, increased awareness campaigns, and comprehensive screening programs, to mitigate the disproportionate impact of tuberculosis within these populations. More research is needed to explore the intricate interplay of these factors and their role in shaping the gender-based disparities in tuberculosis prevalence, ultimately facilitating the development of more effective and tailored public health strategies to curb the tuberculosis burden across diverse demographic groups.

## Conclusion

In conclusion, the retrospective study conducted at the Ho Teaching Hospital provides crucial insights into the prevalence of laboratory-confirmed tuberculosis and newly diagnosed RR-TB within its catchment area. The overall prevalence of laboratory-confirmed tuberculosis was 10.76%, and rifampicin resistance, 3.45%, indicating high infection and possible treatment failure. Considering its infectious nature, this calls for concerted efforts in curbing the spread of the infection.

## Limitation of study

The retrospective design study did not allow the authors to correlate the laboratory findings to the patient clinical presentations.

## Supporting information

**S1 Data.**
(XLSX)

## Acknowledgments

The authors are grateful to the management of the Ho Teaching Hospital and the staff of the laboratory department for their support.

## Author Contributions

**Conceptualization:** John Gameli Deku, Enoch Aninagyei, Israel Bedzina, Prosper Mensah, Solomon Wireko, Innocent Afeke.

**Data curation:** Israel Bedzina, Gameli Nudo, Emmanuel Ativi, Prosper Mensah, Aaron Osei-Tutu, Emmanuel Duker.

**Formal analysis:** Israel Bedzina.

**Investigation:** John Gameli Deku, Gameli Nudo, Emmanuel Ativi.

**Methodology:** John Gameli Deku, Gameli Nudo, Emmanuel Ativi, Aaron Osei-Tutu, Emmanuel Duker.

**Project administration:** John Gameli Deku.

**Resources:** John Gameli Deku, Israel Bedzina, Emmanuel Ativi, Solomon Wireko, Aaron Osei-Tutu, Emmanuel Duker.

**Supervision:** John Gameli Deku, Prosper Mensah, Innocent Afeke.

**Writing – original draft:** John Gameli Deku, Enoch Aninagyei, Israel Bedzina, Solomon Wireko, Innocent Afeke.

**Writing – review & editing:** John Gameli Deku, Enoch Aninagyei, Israel Bedzina, Innocent Afeke.

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
