## [Decision Letter · Decision Letter 0]

29 Dec 2023

PONE-D-23-36410Detection of Mycobacterium tuberculosis and Rifampicin Resistance at the Ho Teaching Hospital of GhanaPLOS ONE

Dear Dr. Deku,

Thank you for submitting your manuscript to PLOS ONE. After careful consideration, we feel that it has merit but does not fully meet PLOS ONE’s publication criteria as it currently stands. Therefore, we invite you to submit a revised version of the manuscript that addresses the points raised during the review process.

We look forward to receiving your revised manuscript.

Kind regards,

Balew Arega Negatie, Msc,MD

Academic Editor

PLOS ONE

2. We note that your Data Availability Statement is currently as follows: All relevant data will be uploaded as a supporting file if the manuscript is accepted for publication

3. We note that Figure 1 in your submission contain satellite images which may be copyrighted. All PLOS content is published under the Creative Commons Attribution License (CC BY 4.0), which means that the manuscript, images, and Supporting Information files will be freely available online, and any third party is permitted to access, download, copy, distribute, and use these materials in any way, even commercially, with proper attribution. For these reasons, we cannot publish previously copyrighted maps or satellite images created using proprietary data, such as Google software (Google Maps, Street View, and Earth). For more information, see our copyright guidelines: http://journals.plos.org/plosone/s/licenses-and-copyright.

We require you to either present written permission from the copyright holder to publish these figures specifically under the CC BY 4.0 license, or remove the figures from your submission:

Reviewers' comments:

Reviewer's Responses to Questions

**Comments to the Author**

1. Is the manuscript technically sound, and do the data support the conclusions?

Reviewer #1: Partly

Reviewer #2: No

Reviewer #3: Yes

2. Has the statistical analysis been performed appropriately and rigorously? 

Reviewer #1: I Don't Know

Reviewer #2: No

Reviewer #3: Yes

3. Have the authors made all data underlying the findings in their manuscript fully available?

Reviewer #1: Yes

Reviewer #2: No

Reviewer #3: Yes

4. Is the manuscript presented in an intelligible fashion and written in standard English?

Reviewer #1: Yes

Reviewer #2: No

Reviewer #3: Yes

5. Review Comments to the Author

Reviewer #1: All comments have been incorporated in the marked manuscript. For example, the introduction is silent on the subject matter which was supposed to have been studied. For example, there are several studies on Rifampicin resistance in Ghana but the authors rarely mentioned any of these.

Reviewer #2: Dear authors

1. Title

The title should be revised as “Trends of Mycobacterium tuberculosis and rifampicin resistance……”

2. Abstract section

a. Lacks research gaps

b. Its results were not support with statistical values and mix-up with recommendations

c. Its conclusion is not well stated

3. Introduction

a. Need a through language edition

b. Lack coherence

c. Inconsistence citation (line 96)

4. Method section

a. Study area description lacks coherence

b. No criteria for inclusion/exclusion of data type/participants

c. Data collection not well described; what variables were collected

d. Variables (outcome and independents variables) are missing

5. Results

a. This section should be reanalysis; the way you wrote AOR? Not to the standard

6. Discussion

a. You reviewed very few literature for the discussion. Look at line number 253 to 306, you didn’t use any literature.

b. Overall you use 15 paper to prepare this manuscript

7. Conclusion

a. Your conclusion should be based on major findings; which is missed in this case

Reviewer #3: The manuscript is well written and method used appropriate to generate the data presented and analyzed. Although there is a concern on the results presenting data on rifampicin susceptibility testing (table as sensitive-Intermediate-Resistant) in table 2 (Table 2 Prevalence and risk factors of newly diagnosed rifampicin resistant tuberculosis among suspected

174 cases at the Ho Teaching Hospital between 2016 and 2021). The method used talked of results of Gene Xpert method and not the cultural method.

There is also a limitation using this retrospective method which did not enable the authors to correlate the laboratory findings to the clinical presentations.

In conclusion the study site was appropriate to determine the epidemiological trends given its high turn around and the number of suspicious of cases of M. tuberculosis per year. The data generated well appropriately analyzed and the conclusions are adequate to the study design.

6. PLOS authors have the option to publish the peer review history of their article (what does this mean?). If published, this will include your full peer review and any attached files.

Reviewer #1: No

Reviewer #2: No

Reviewer #3: **Yes: **

---

## [Author Response · Author response to Decision Letter 0]

25 Jan 2024

University of Health and Allied Sciences

PMB 31

Ho

January 25, 2024

The Editor

Plos One

Dear Editor,

RESPONSE TO REVIEWERS COMMENT

The authors of the manuscript “Trends of Mycobacterium tuberculosis and rifampicin resistance at the Ho Teaching Hospital in Ghana.”, with manuscript ID. PONE-D-23-36410 are grateful for your mail. We are equally thankful for the time you spent on the manuscript to enrich it for possible publication in your esteemed journal.

We write to respond to your comments as follows:

Reviewer #1: 

Comment: “All comments have been incorporated in the marked manuscript. 

For example, the introduction is silent on the subject matter which was supposed to have been studied. For example, there are several studies on Rifampicin resistance in Ghana but the authors rarely mentioned any of these”.

Authors’ response: All corrections have been made as suggested in the manuscript. However, the authors wish to clarify that, the months under the result section are cumulative over the six-year period. (lines 574-580)

Reviewer#2: 

1. Title

Reviewer comment: “The title should be revised as “Trends of Mycobacterium tuberculosis and rifampicin resistance……” 

Authors’ response: The title has been revised to “Trends of Mycobacterium tuberculosis and rifampicin resistance at the Ho Teaching Hospital in Ghana”. See lines 1 and 2.

2. Abstract section

Reviewer comment: “a. Lacks research gaps”

Authors’ response: The abstract has been revised to include the research gap. See lines 36-37.

Reviewer comment: “b. Its results were not support with statistical values and mix-up with recommendations”

Authors’ response: Authors have added statistical values (Lines 51 and 78) and removed recommendations.

Reviewer comment: “c. Its conclusion is not well stated”

Authors’ response: The conclusion has been restated. See lines 82-84.

Reviewer comment 3. Introduction

Reviewer comment: “a. Need a through language edition”

Authors response: Language editing was done as suggested.

Reviewer comment b “Inconsistence citation (line 96)”

Authors’ response: “All comments have been addressed. 

4. Method section

Reviewer comment: “a. Study area description lacks coherence”

Authors’ response: This section has been rephrased to improve coherence. 

Reviewer comment: “b. No criteria for inclusion/exclusion of data type/participants”

Authors’ response: The inclusion and exclusion criteria section for data type/participants have been added. See 239-242

Reviewer comment: “c. Data collection not well described; what variables were collected”

Authors’ response: Variables obtained for the study have been added to the data collection section. See line 361-367. 

Reviewer comment: “d. Variables (outcome and independents variables) are missing”

Authors response: Variables (outcome and independents variables) session have been added. See line 361-367.

5. Results

Reviewer comment a. “This section should be reanalysis; the way you wrote AOR? Not to the standard”

Authors response: Authors are unable to work on this comment due to a lack of clarity on what is expected from us. However, the meaning of the abbreviations used in the table has been added as a footnote. See Table 1

6. Discussion

Reviewer comment: “a. You reviewed very few literatures for the discussion. Look at line number 253 to 306, you didn’t use any literature”.

“b. Overall you use 15 paper to prepare this manuscript”

Authors response: Comparative analysis has been done and more literatures cited. 

7. Conclusion

Reviewer comment: “a. Your conclusion should be based on major findings; which is missed in this case”.

Authors’ response: Authors have revised study conclusions. See line 1010-1015

Reviewer #3:

Reviewer comment: “The manuscript is well written and method used appropriate to generate the data presented and analyzed. Although there is a concern on the results presenting data on rifampicin susceptibility testing (table as sensitive-Intermediate-Resistant) in table 2 (Table 2 Prevalence and risk factors of newly diagnosed rifampicin resistant tuberculosis among suspected 174 cases at the Ho Teaching Hospital between 2016 and 2021). The method used talked of results of Gene Xpert method and not the cultural method”.

Authors response: The Gene Xpert machine used reports rifampicin susceptibility results as “Susceptible”, “Indeterminate”, or “Resistant” where Indeterminate means neither “Susceptible” nor “Resistant” which could be due to insufficient bacterial load.

Reviewer comment: “There is also a limitation using this retrospective method which did not enable the authors to correlate the laboratory findings to the clinical presentations.”

Authors response: Suggested limitation of study has been added. See lines 1031- 1033

Reviewer comment: “In conclusion the study site was appropriate to determine the epidemiological trends given its high turn around and the number of suspicious of cases of M. tuberculosis per year. The data generated well appropriately analyzed and the conclusions are adequate to the study design”.

Authors’ response: Thank you.

Thank you.

…Signed……..

John Gameli Deku

(Lead and corresponding author)

---

## [Decision Letter · Decision Letter 1]

2 Apr 2024

PONE-D-23-36410R1Trends of Mycobacterium tuberculosis and rifampicin resistance at the Ho Teaching Hospital in Ghana.PLOS ONE

Dear Dr. Deku,

Thank you for submitting your manuscript to PLOS ONE. After careful consideration, we feel that it has merit but does not fully meet PLOS ONE’s publication criteria as it currently stands. Therefore, we invite you to submit a revised version of the manuscript that addresses the points raised during the review process. Please submit your revised manuscript by May 17 2024 11:59PM. If you will need more time than this to complete your revisions, please reply to this message or contact the journal office at plosone@plos.org. Please include the following items when submitting your revised manuscript:A rebuttal letter that responds to each point raised by the academic editor and reviewer(s). You should upload this letter as a separate file labeled 'Response to Reviewers'.A marked-up copy of your manuscript that highlights changes made to the original version. You should upload this as a separate file labeled 'Revised Manuscript with Track Changes'.An unmarked version of your revised paper without tracked changes. You should upload this as a separate file labeled 'Manuscript'.

We look forward to receiving your revised manuscript.

Kind regards,

Balew Arega Negatie, Msc,MD

Academic Editor

PLOS ONE

Journal Requirements:

Reviewers' comments:

Reviewer's Responses to Questions

**Comments to the Author**

1. If the authors have adequately addressed your comments raised in a previous round of review and you feel that this manuscript is now acceptable for publication, you may indicate that here to bypass the “Comments to the Author” section, enter your conflict of interest statement in the “Confidential to Editor” section, and submit your "Accept" recommendation.

Reviewer #1: (No Response)

2. Is the manuscript technically sound, and do the data support the conclusions?

Reviewer #1: Yes

3. Has the statistical analysis been performed appropriately and rigorously? 

Reviewer #1: Yes

4. Have the authors made all data underlying the findings in their manuscript fully available?

Reviewer #1: Yes

5. Is the manuscript presented in an intelligible fashion and written in standard English?

Reviewer #1: Yes

6. Review Comments to the Author

**Reviewer #1:** I see a massive improvement in the manuscript. It needs further revisions though. Well done to the authors.

7. PLOS authors have the option to publish the peer review history of their article (what does this mean?). If published, this will include your full peer review and any attached files.

Reviewer #1: No

---

## [Author Response · Author response to Decision Letter 1]

17 May 2024

University of Health and Allied Sciences

PMB 31

Ho

May 09, 2024

The Editor-in-Chief

Plos One

Dear Editor-in-Chief,

RESPONSE TO COMMENTS

The authors of the manuscript “Trends of Mycobacterium tuberculosis and rifampicin resistance at the Ho Teaching Hospital in Ghana.”, with manuscript ID. PONE-D-23-36410R1 are grateful for your mail.

We wish to state that there were no comments from the Editor or the Reviewers for us to address.

However, on the request from the editorial office titled “Journal Requirements”, we wish to respond as follows:

1. The reference list has been reviewed. However, no change was made.

2. No cited paper was retracted.

Due to the aforementioned reasons, the previous version of the manuscript has been maintained.

Please, do not hesitate to contact us with any other concern(s).

Thank you.

…Signed……..

John Gameli Deku

(Lead and corresponding author)

---

## [Editor Report · Decision Letter 2]

27 May 2024

Trends of Mycobacterium tuberculosis and rifampicin resistance at the Ho Teaching Hospital in Ghana .

PONE-D-23-36410R2

Dear Dr. John Gameli 

We’re pleased to inform you that your manuscript has been judged scientifically suitable for publication and will be formally accepted for publication once it meets all outstanding technical requirements.

Kind regards,

Balew Arega Negatie, Msc,MD

Academic Editor

PLOS ONE
---

## [Editor Report · Acceptance letter]

29 May 2024

PONE-D-23-36410R2 

PLOS ONE

Dear Dr. Deku, 

I'm pleased to inform you that your manuscript has been deemed suitable for publication in PLOS ONE. Congratulations! Your manuscript is now being handed over to our production team.

Kind regards, 

on behalf of

Dr. Balew Arega Negatie 

Academic Editor

PLOS ONE